# The Fungus *Beauveria bassiana* Alters Amounts of Sterols, Fatty Acids, and Hydroxycinnamic Acids in Potato *Solanum tuberosum*

**DOI:** 10.3390/plants12233938

**Published:** 2023-11-22

**Authors:** Maksim Tyurin, Elena Chernyak, Oksana Tomilova, Khristina Tolokonnikova, Svetlana M. Malysh, Elena Khramova, Sergey Morozov, Vadim Kryukov

**Affiliations:** 1Institute of Systematics and Ecology of Animals, Siberian Branch of Russian Academy of Sciences (SB RAS), Frunze Str. 11, Novosibirsk 630091, Russia; maktolt@mail.ru (M.T.); toksina@mail.ru (O.T.); klatty@yandex.ru (K.T.); 2N.N. Vorozhtsov Novosibirsk Institute of Organic Chemistry, SB RAS, Academician Lavrentyev Ave. 9, Novosibirsk 630090, Russia; chernyak@nioch.nsc.ru (E.C.); morozov@nioch.nsc.ru (S.M.); 3All-Russian Institute of Plant Protection, Podbel’skogo Str. 3, St. Petersburg 196608, Russia; s.malysh-vizr@yandex.ru; 4Central Siberian Botanical Garden SB RAS, Zolotodolinskaya Str. 101, Novosibirsk 630090, Russia; elenakhramova2023@yandex.ru

**Keywords:** endophyte, entomopathogen, lipids, phenolic acids, fatty acids, stigmasterol

## Abstract

The entomopathogenic endophytic fungus *Beauveria bassiana* can colonize plants resulting in growth promotion and protection against phytopathogenic microorganisms. However, physiological changes in potato plants (*Solanum tuberosum*) during this interaction are poorly understood. In the present work, gas chromatography–mass spectrometry and high-performance liquid chromatography were used to analyze sterol, fatty acid, and phenolic acid concentrations in potato plants inoculated with *B. bassiana* conidia in soil. We showed an increase in amounts of stigmasterol, minor sterol compounds, and some hydroxy fatty acids in leaves after the fungal treatment. Moreover, levels of hydroxycinnamic acids, especially chlorogenic acid, were elevated in roots following the *B. bassiana* inoculation. We propose that these changes could have been caused by oxidative reactions, and the alterations may have resulted in growth-stimulatory and protective effects of *B. bassiana* on the plants.

## 1. Introduction

Entomopathogenic endophytic fungi such as *Beauveria* and *Metarhizium* are promising candidates for improving plant health and yield [1,2,3]. Traditionally, products for the management of herbivores and blood-sucking arthropods have been developed based on their conidia and blastospores. Broad host ranges and the ability to penetrate the cuticle of arthropods give the fungi an advantage over bacterial and viral biocontrol agents. In the 21st century, it was discovered that these ascomycetes lead a dual lifestyle: aside from parasitism on arthropods, the fungi are facultative symbionts of plants [2]. These fungi are able to colonize roots and aboveground plant organs, thereby stimulating plant growth and immunity and protecting plants against phytopathogens and herbivorous insects [1,2]. Growth- and immune-system-modulating actions of fungi on plants are mediated by several mechanisms including increased nutrient absorption by roots and production of secondary metabolites and hormones eliciting the differential regulation of metabolic and immune signaling pathways [4,5,6,7]. For example, Raad et al. [6] showed that colonization of *Arabidopsis* by *Beauveria* led to reprogramming of metabolic pathways related to resistance to environmental stresses and phytopathogenic microorganisms. Proietti et al. [7] reported the differential regulation of proteins associated with defense responses, calcium transport, protein/amino acid turnover, and biosynthesis of energy compounds and hormones in tomato in response to colonization with *B. bassiana*. Entomopathogenic fungi are able to alter the biochemical composition of plant tissues [8] and the attractiveness of plants to phytophagous insects [9].

Several important effects have been found during interaction of entomopathogenic fungi (*Beauveria* or *Metarhizium*) with potato *Solanum tuberosum* plants. Rios-Moreno et al. [10] demonstrated the ability of *Metarhizium brunneum* to colonize potato tissues and produce destruxin under laboratory conditions after spraying the plants with conidial suspensions. Krell et al. [11,12] showed in laboratory experiments that *M. brunneum* is capable of endophytic colonization of potato plants, thus increasing dry biomass, nitrogen, and phosphorus contents of the plants, as well as improving water use efficiency. Changes in these parameters proved to be mediated by different nutrient conditions, and these authors stated that *M. brunneum* can mitigate nutrient deficits in the soil, further improving plant productivity [12]. In a hydroponic condition, inoculation of a culture medium with fungi *Beauveria bassiana* or *Metarhizium robertsii* was found to alter pigment ratios, peroxidase activity, levels of thiobarbituric acid reactive substances, and proline in various tissues of potato [13]. The authors hypothesized that the plants exhibit symptoms of moderate oxidative stress during colonization by the fungi [13]. It was also shown that under the conditions of a natural load of *Beauveria* and *Metarhizium* propagules in the soil of potato fields (10^2^–10^4^ CFU/g), the occurrence of *Metarhizium* in plants without surface sterilization may reach 50%, but the colonization of internal potato tissues occurs rarely [14]. Nevertheless, the preplanting treatment of potato tubers with either *B. bassiana* or *M. robertsii* reduced the incidence of rhizoctoniosis (*Rhizoctonia solani*), changed immune-activity parameters, and improved crop quality under field conditions [15]. In particular, *B. bassiana* (strain Sar-31) significantly decreased Rhizoctonia disease and improved the tuber yield [15].

It is known that interactions of plants with arbuscular mycorrhizal fungi cause alterations in the profile of plant sterols, fatty acids, terpenoids, and other compounds [16,17,18,19], and these changes may be accompanied by alterations in resistance to pathogens and herbivores. Plants produce a mixture of sterols including sitosterol, campesterol, stigmasterol, and cholesterol [20]. These compounds are involved in the maintenance of the fluidity and permeability of the plasma membrane in plant cells and in signaling and stress responses [20,21]. Moreover, sterol production has an important role in plant immunity against phytopathogens [22]. Fatty acids are ingredients of cellular membranes, constituents of the extracellular barrier, and they are important as carbon and energy reserves and in the salicylate and jasmonate signaling pathways; therefore, these acids participate in plant responses to abiotic and biotic stresses [23]. Changes in lipid composition of plants during colonization by *Beauveria* or *Metarhizium* are poorly studied. Nonetheless, it has been shown by transcriptomic and proteomic approaches that colonization of plants by these fungi causes differential regulation of genes or proteins related to lipid metabolism [6,7], including lipid peroxidation [7]. Studies on potato lipid profiles have mainly been focused on the nutritional quality of tubers (e.g., [24]). To our knowledge, there have been no studies on the influence of fungi on the lipid composition of roots and shoots.

Interactions of plants with fungi may change the amounts of phenolic compounds [25]. These compounds in plants are represented by monomers, oligomers, and polymers. Flavonoids and phenylpropanoids, in particular hydroxycinnamic acids, are the largest class of phenolic compounds. Functional roles of phenolic compounds in plants are diverse, including inactivation of reactive oxygen species (ROS) and consequent slowing down of the oxidation of cell membrane lipids [26]. Hydroxycinnamic acids, in particular chlorogenic acid and its isomers, are the major phenolic compounds of potato [27]. It is known that chlorogenic acid is involved in the regulation of plant growth and development, e.g., it has a positive impact on the development of lateral roots [28,29,30]. Inoculation of plants with endophytic fungi can promote the production of chlorogenic acid in roots, as shown, for example, in a *Chaetomium*–*Miscanthus sinensis* system [25]. Some endophytic microorganisms such as *Bacillus subtilis* can themselves produce chlorogenic acid and its derivatives [31]. This acid possesses antioxidant activity and plays an important role in plants’ defensive reactions to phytopathogenic microbes [32,33] and nematodes [34].

We have previously observed growth stimulation, increases in levels of antioxidant enzymes, and enhanced resistance to phytopathogens in potato plants exposed to entomopathogenic fungi [13,15]. Therefore, here we hypothesized that the colonization can lead to alterations in the lipid and phenylpropanoid profiles of roots and aboveground organs. In the present paper, using gas chromatography–mass spectrometry (GC-MS) and high-performance liquid chromatography (HPLC), we analyzed levels of sterols, fatty acids, and hydroxycinnamic acids in roots and leaves of potato after colonization by *B. bassiana* under laboratory conditions.

## 2. Results

### 2.1. Colonization and Growth Parameters

The extent of colonization of internal plant tissues by *B. bassiana* at 14 days postinoculation was 45% for roots, 75% for stems, and 10% for leaves (Figure 1A). No *B. bassiana*-colonized tissues were found in the control treatment group. Cultures reisolated from plants showed 100% identity of the secreted-lipase gene fragment (445 bp) toward each other and to the paternal strain Sar-31 (Appendix A). On the other hand, the fragment was variable enough to discriminate different *B. bassiana* strains, including those originating from the same location, namely, Sar-31 and BBK-1 (Appendix A). Thus, potential contamination of a particular culture of *Beauveria* with a random fungal strain is highly likely to be tracked down by genotyping, showing a different sequence. This allows us to presume that there was no contamination during the study and that all the cultures used for the experiment correspond to the paternal strain Sar-31.

Plants inoculated with *B. bassiana* manifested a 1.3-fold increase in shoot weight as compared to the control (*t* = 2.51, *p* = 0.016, Figure 1B). Root weight increased 1.2-fold in response to the fungal treatment, but the difference was not significant (*t* = 1.65, *p* = 0.11).

### 2.2. Amounts of Sterols and Fatty Acids

In nonpolar extracts of leaves and roots, several phytosterols were identified; among them were stigmasta-3,5-dien, cholesterol, campesterol, stigmasterol, β-sitosterol, and fucosterol (Figure 2A). In response to *B. bassiana* colonization, the total amount of sterols in leaves increased 1.3-fold, but the change was not statistically significant (Z = 1.6, *p* = 0.11 as compared to the control). Nevertheless, a significant 1.5-fold increase in a major compound (stigmasterol) was registered in response to the fungal inoculation (*t* = 4.3, *p* = 0.005, Figure 2B). Moreover, a weak elevation of cholesterol and β-sitosterol levels (1.2-fold, *p* > 0.17) and significant (2-fold, *p* = 0.003) upregulation of a minor compound (stigmasta-3,5-dien) were registered. There were no significant changes in the amounts of sterols in roots in response to *B. bassiana* treatment (Figure 2C).

A total of 8 unsaturated fatty acids (C15–C22) and 17 saturated ones (C9–C28) were detected in potato leaves and roots (Figure 3). The total level and individual concentrations of most of the fatty acids did not change under the influence of *B. bassiana* in both organs. Exceptions were acids C15:0, C15:0 14Me, and α-hydroxy fatty acid C16:0 2-OH, whose content significantly increased in the leaves of colonized plants compared to control ones (Figure 3A,B). In particular, the amount of C15:0 went up by 2.4-fold as compared to the control (Z = 2.4, *p* = 0.02), that of C15:0 14Me by 1.2-fold (Z = 2.1, *p* = 0.04), and the amount of C16:0 2-OH by 1.6-fold (*t* = 4.4, *p* = 0.004). There were no significant effects of *B. bassiana* on any fatty acid amounts in roots (Figure 3C,D). PCA of fatty acid profiles revealed a strong overlap of clasters (Figure 4), indicating only a minor influence of the fungal colonization on the profile of these lipids.

### 2.3. Amounts of Hydroxycinnamic Acids

In extracts of roots, three hydroxycinnamic acids were found: chlorogenic (5-O-caffeoylquinic) acid, caffeic acid, and an unidentified hydroxycinnamic acid (Appendix A). The total amount of hydroxycinnamic acids in roots increased 1.5-fold in response to the colonization by *B. bassiana* (*t* = 3.1, *p* = 0.013). In particular, the amount of chlorogenic acid rose by 1.5-fold (Z = 2.2, *p* = 0.029, Figure 5A). Concentration of the unidentified hydroxycinnamic acid increased by 1.6-fold (*t* = 3.0, *p* = 0.022) and the amount of caffeic acid by 1.1-fold (*p* = 0.40) as compared to the control.

In extracts of leaves, two hydroxycinnamic acids were detected: chlorogenic (5-O-caffeoylquinic) acid and cryptochlorogenic (4-O-caffeoylquinic) acid (Appendix A). No significant differences in the amounts of these compounds were noted in leaves in response to the *B. bassiana* treatment (Figure 5B).

## 3. Discussion

The ability of *B. bassiana* to colonize potato and to increase plant biomass is consistent with our previous studies [15]. In the present work, we registered a significant elevation in weight of aboveground organs and a minor increase in root weight 14 days after inoculation of the soil with fungal conidia (Figure 1). Several mechanisms can mediate growth stimulation, as shown for different entomopathogenic fungus–plant systems [4,5,35,36]; in particular, for potato, fungi are known to improve nutrient absorption and water use efficiency [11,12]. Furthermore, growth stimulation may be caused by changes in the production of hydroxycinnamic acids (Figure 4) following fungal colonization.

We observed enhancement of stigmasterol production in leaves of potato plants inoculated with *B. bassiana* (Figure 2). Stigmasterol is synthesized from β-sitosterol by desaturation: the introduction of a double bond at the C22 position [37]. Griebel and Zeier [38] showed an upregulation of sterol desaturase CYP710A1 and stigmasterol production in *Arabidopsis* in response to infection with biotrophic (*Pseudomonas syringae*) and necrotrophic (*Botrytis cinerea*) pathogens. Higher expression of sterol desaturase has also been detected after infection of *Arabidopsis* with biotrophic and hemibiotrophic fungi *Golovinomyces cichoracearum* and *Phytophthora infestans* [39,40]. Griebel and Zeier [38] report that the elevation of stigmasterol is triggered by pathogen-associated molecular patterns and ROS but does not depend on jasmonic acid, salicylic acid, or ethylene defense pathways. Our previous papers also indicate moderate oxidative stress in potato after *B. bassiana* colonization, as evidenced by upregulation of malondialdehyde and antioxidant enzymes [13,15]. An increase in ROS production in tomato tissues in response to inoculation with *B. bassiana*, *Metarhizium brunneum*, or *Trichoderma harzianum* was observed by Gupta et al. [41]. Raad et al. [6] showed that two *B. bassiana* strains induce genes involved in oxidative stress processes in *Arabidopsis thaliana*, also in agreement with results on “Solanaceae plant–entomopathogenic fungus *B. bassiana*” systems [13,41]. Pose et al. [42] demonstrated an essential role of sterols in the localization of NADPH oxidases for the regulation of ROS in plants. Therefore, we propose that upregulation of sterol production and especially stigmasterol may be considered a response of potato to the oxidative stress caused by *B. bassiana*. Notably, the increased concentration of the stigmasterol in *Arabidopsis* after *P. syringae* infection did not lead to the downregulation of β-sitosterol [38], consistent with our findings and pointing to the de novo synthesis of β-sitosterol following colonization by *B. bassiana*.

Interestingly, Griebel and Zeier [38] found that an increase in the stigmasterol level in plants promotes the proliferation of the biotrophic pathogen *P. syringae* but not the necrotrophic pathogen *B. cinerea*. Those authors suggested that stigmasterol can interfere with a specific plant defense pathway (e.g., flavin-dependent monooxygenases) or act on microorganisms directly by enhancing their growth. The behavior of *B. bassiana* in a plant is biotrophic according to the classification of Spanu and Kamper [43] because colonization does not induce drastic changes in plant immunity and because tissues remain alive during fungal growth [2]. It is possible that elevation of the stigmasterol is favorable for colonization of plant tissues by the fungus; nevertheless, further experimentation is required to uncover the underlying mechanisms.

We did not observe any serious alteration in the fatty acid composition of potato roots and leaves after the treatment with *B. bassiana*. Previous works showed changes in the profile of fatty acids caused by arbuscular mycorrhiza. For instance, Cao et al. [44] demonstrated the upregulation of saturated fatty acids and downregulation of unsaturated fatty acids after the colonization of trifoliate orange *Poncirus trifoliata* plants by *Piriformospora indica*. Lida et al. [45] showed the differential regulation of genes involved in fatty acid biosynthesis in cucumber after the spraying of leaves with *B. bassiana*. We detected statistically significant quantitative changes only for the third type of fatty acids in their broad spectrum: C15:0, C15:0 14Me, and C16:0 2-OH (Figure 3). It is unclear why the quantitative changes affected these compounds. Probably, the elevation in the C16:0 2-OH amount was caused by the enhancement of lipid peroxidation in the plants. In particular, Proietti et al. [7] showed that treatment of tomato with *B. bassiana* causes the differential regulation of genes related to these processes. Notably, hydroxy fatty acids are characterized by higher viscosity and reactivity as compared to other fatty acids and possess the strongest fungicidal properties [46]. This finding is in agreement with the greater resistance of potato to phytopathogenic fungi after colonization by *B. bassiana* [15].

We observed an elevated amount of hydroxycinnamic acids, especially chlorogenic acid, in potato roots after *B. bassiana* inoculation (Figure 5). It was shown in a recent work by Muola et al. [47] that *B. bassiana* colonization of oilseed rape *Brassica napus* induces the biosynthesis of flavonoids including hydroxycinnamic acids and chlorogenic acid. A significant increase in flavonoid concentration in cucumber plants during colonization with *B. bassiana* (28 days postinoculation) was also detected by Homayoonzadeh et al. [48]. Researchers [47] hypothesized that mechanisms of this induction may involve a defense response of the plant to *B. bassiana* or fungal biosynthesis of some compounds; this issue should be elucidated in future studies. It is well known that chlorogenic acid is an antioxidant; therefore, this phenomenon may be associated with a response of potato to oxidative processes. Mei et al. [49] demonstrated that exogenous chlorogenic acid diminishes the levels of H_2_O_2_ and malondialdehyde in apple leaves treated with an herbicide inducing oxidative stress. Moreover, exogenous chlorogenic acid modulates peroxidase and catalase production and the expression regulation of genes related to the antioxidant system and phenolic compound metabolism [49]. Those authors believe that chlorogenic acid attenuates membrane damage and lipid oxidation in plants.

Chlorogenic acid plays an important role in processes of plant growth and organogenesis of shoots and roots and also affects fruit ripening [30,50,51]. The effect of chlorogenic acid may be dose-dependent. In particular, a low concentration of chlorogenic acid accelerates growth of pear tree *Pyrus betulaefolia*, and a high concentration of the acid can inhibit it [50]. The concentration of chlorogenic acid is closely linked with the amount of a growth-stimulating compound (an auxin called indole-3-acetic acid). This mechanism is mediated by inhibition of indole-3-acetic acid oxidase by phenolic acids, including chlorogenic acid [52,53]. Therefore, the upregulation of chlorogenic acid may be one of the reasons for the growth stimulation seen after colonization by *B. bassiana*. Chlorogenic acid has antifungal properties, in particular, the inhibition of spore germination, a reduction in mycelial growth, or cell lysis, which have been documented for *Sclerotinia sclerotiorum, Fusarium solani*, *Verticillium dahliae*, *Botrytis cinerea*, *Cercospora sojina* [33], *Phytophthora infestans* [54], and *Phlyctaena vagabunda* [55]. This acid causes membrane permeabilization and disruption of the membrane structure, as shown with *F. solani* and *Candida albicans* [33,56]. Moreover, chlorogenic acid is an important intermediate in lignin biosynthesis, and lignification may be one of the key mechanisms of disease resistance [32]. Thus, the elevation of the chlorogenic acid level in potato after the *B. bassiana* treatment is in agreement with diminished incidence of diseases caused by phytopathogenic fungi [15].

## 4. Material and Methods

### 4.1. The Fungus and Its Cultivation

An isolate of *Beauveria bassiana* s.s. (strain Sar-31) from the collection of microorganisms at the Institute of Systematics and Ecology of Animals (the Siberian Branch of the Russian Academy of Sciences) was used. The strain was isolated from *Calliptamus italicus* (Orthoptera, Acrididae) in Western Siberia (53°41′ N 78°02′ E) in 2001. The fungal species was identified on the basis of the sequence of a translation elongation factor region (GenBank accession No. MZ564259). The fungus was grown in Petri dishes (90 mm) with 1/4 Sabouraud dextrose agar supplemented with 0.25% of yeast extract (1/4 SDAY [57]) at 26 °C in the dark for 10 days. For the treatment of plants, conidia were harvested from agar by means of a spatula and dispersed in an aqueous solution of Tween 80 (0.03% *v*/*v*). The concentration of conidia was determined using a hemocytometer. The germination of conidia on 1/4 SDAY was 98% after 16 h of incubation (n = 100).

### 4.2. Plant Cultivation and Inoculation with the Fungus

Potato plants (*Solanum tuberosum* variety Agata) were used in the experiments. Tubers (diameter 3–4 cm) were treated with 7% sodium hypochlorite (2 min) and then 70% ethanol, followed by three rinses in sterile distilled water [12]. The tubers were pre-exposed to light at 25 °C for 30 days for initiation of germination (shoot length ~3 mm) [12]. Germinated tubers were planted individually in cone-shaped containers (1 L, height: 12 cm, lower diagonal: 10 cm, upper diagonal: 14 cm) with 300 g of unsterilized soil substrate Terra Vita (Nord Palp, Fornosovo, Russia). The substrate includes peat, purified sand, perlite, mineral micro- and macro-elements, and has a pH of 6.5. The substrate was analyzed for the presence of fungi *Metarhizium* and *Beauveria* by plating on CTAB medium [58,59] with a minor modification [14,15], and these fungi were not detectable.

Plants were grown at 22 °C and 40–60% relative humidity with a 16 h/8 h photoperiod (light/dark) using full-spectrum Flora phytolamps (Osram, Munich, Germany). We used soil drenching with the conidial suspension around seedlings ([60,61,62] with modifications) because preliminary assays showed effective colonization of potatoes under the laboratory conditions described above. Soil around tubers was inoculated with 10 mL of the suspension (10^8^ conidia per mL) in the 0.03% (*v*/*v*) Tween 80 solution 3 days after planting. Control plants were treated with a conidia-free aqueous Tween 80 solution.

### 4.3. The Extent of Colonization and Plant Weight

The time point “14 days after treatment” was selected for analysis. This choice is due to preliminary assays showing that this period is sufficient for implementing the colonization of plant tissues by the fungal strain. Plants were excavated and carefully cleansed from the soil substrate. Fresh weights of roots and shoots were determined with a precision of 0.01 g. Endophytic colonization of the plants was estimated according to Posada et al. [63]. Roots, stems, and leaves were surface-sterilized by means of 0.5% sodium hypochlorite for 2 min and 70% ethanol for 2 min with subsequent 3-fold washing in sterile distilled water. Then, the plant organs were plated in 90 mm Petri dishes (two fragments of a root or stem or two leaves per dish) on the CTAB medium [58,59] with a modification (10 g/L peptone, 40 g/L D-glucose, 20 g/L agar, 1 g/L yeast extract, 0.35 g/L cetyltrimethylammonium bromide, 0.05 g/L cycloheximide, 0.05 g/L tetracycline, and 0.6 g/L streptomycin). For verification of surface sterilization, tissue imprinting onto this medium was performed [64]. The dishes were incubated at 24 °C for 12 days, and fungal growth was examined visually. The colonies were identified as *Beauveria* by light microscopy. Twenty plants per treatment group were used for the determination of weight and colonization frequency.

### 4.4. PCR-Based Confirmation of Reisolate Identity

To confirm the identity of endophytic fungi reisolated from plants exposed to the initial *B. bassiana* strain Sar-31, genomic DNA extraction was followed by PCR with specific primers and amplicon purification and sequencing. For the precise discrimination of the fungal strain, which may have homologous sequences of a translation elongation factor and other widely used genes [65,66], an approach of Levchenko et al. [67] was applied, utilizing another, more variable locus. Primers slipF1 (5′-GTGGAAGCTCGCCGAGAG-3′) and slipR1 (5′-GCAGATGSACCTCGTTGC-3′) were used, which flank a ~500 bp region of the gene of a secreted lipase: a protein from the superfamily of a/β-hydrolases. Five additional strains of *B. bassiana* from the collection of microorganisms at the All-Russian Institute of Plant Protection were chosen for sequence variation comparison (Appendix A).

A simplified protocol of DNA extraction [68] was applied to the samples of fungal mycelia grown in pure culture. PCR products were separated by gel electrophoresis, and bands of specific size were excised, gel-purified according to Vogelstein and Gillespie [69], and sequenced in both directions on an ABI Prism sequencer (Thermo Fisher Scientific, Waltham, MA, USA). The obtained sequences were corrected manually and aligned in BioEdit [70].

### 4.5. Plant Material for Lipid and Phenolic Acid Quantification

Fourteen days after fungal treatment, plant roots and leaves were collected, lyophilized in individual packs, and stored at −20 °C. In the *B. bassiana* treatment group, only plants that exhibited the colonization of roots and stems were chosen for the quantitation of lipids and phenolic acids. In a control treatment group, plants that did not exhibit fungal colonization were selected for the above analyses. Each sample consisted of four pooled plants, and four samples (biological replicates) were used for the determination of lipid and hydroxycinnamic acid amounts.

### 4.6. Extraction of Lipid Compounds and Derivatization

The dry plant material was crushed in a ball mill and extracted with a mixture of chloroform and methanol by the Folch method [71,72] in an ultrasonic bath at 25 °C for 15 min. Water was added to the supernatant, it was centrifuged for 15 min at 2200 g, and the organic layer was separated and evaporated at 40 °C in a water bath. The lipid extract was derivatized with 1% H_2_SO_4_ in MeOH at 80 °C for 3 h [73,74] and then analyzed by gas chromatography–mass spectrometry (GC/MS).

### 4.7. Extraction of Phenolic Acids

The plant material was crushed in the ball mill, and phenolic acids were extracted with a mixture of methanol and 2% aqueous acetic acid (2:3 *v*/*v*) [75]. The material was extracted in the ultrasonic bath at 50 °C for 30 min, followed by centrifugation and decanting. The solution was subjected to HPLC.

### 4.8. GC/MS and HPLC Analyses

GC/MS analysis was performed with a 6890 N gas chromatograph and a 5975 N mass-selective detector (Agilent Technologies, Santa Clara, CA, USA). Separation was carried out on an HP-5MS capillary column (30 m × 0.25 mm, film thickness 0.25 μm), at an injector temperature of 280 °C, ion source temperature of 230 °C, quadrupole temperature of 150 °C, and a flow rate of a carrier gas (helium) of 1 mL/min. A temperature gradient was implemented as follows: the temperature was initially held at 50 °C for 2 min and then raised to 280 °C at a rate of 10 °C/min. The ChemStation software, the NIST 14 MS spectral library, and literature data [76,77,78] were used for identifying fatty acid methyl esters and sterols. Methyl stearate and octacosane (Sigma-Aldrich, St. Louis, MO, USA) were employed for the quantification of fatty acid methyl esters and sterols.

HPLC analysis was performed on an Agilent LC 1100 chromatograph (Agilent Technologies, Santa Clara, CA, USA) equipped with a quaternary pump, an autosampler, and a diode array detector. The chromatographic conditions were as follows: a ZORBAX Eclipse XBD-C8 column (4.6 mm × 150 mm, 5 μm) (Agilent Technologies); a mobile phase consisting of methanol/0.1% (*v*/*v*) trifluoroacetic acid in an H_2_O gradient; the methanol percentage in the gradient was 2–100% (minutes 0–20) and 100% methanol (minutes 20–25); the flow rate was 0.8 mL/min; the injection volume was 2 μL; and the detection was performed simultaneously at four wavelengths: 254, 280, 320, and 360 nm. Chlorogenic acid (Sigma-Aldrich) was used for the quantification of hydroxycinnamic acids. Identification of cryptochlorogenic and caffeic acids was carried out on the basis of UV spectra and retention times of peaks [79].

### 4.9. Statistics

Data were checked for normality by the Shapiro–Wilk *W* test and for equality of variances by Levene’s test [80]. Normally distributed data were analyzed by the *t* test, and non-normally distributed data by the Mann–Whitney *U* test. Plots are presented as the mean and standard error. Principal component analysis (PCA) of fatty acid composition was performed using a variance–covariance matrix. The Past v.3 software [81] was utilized for all the analyses.

## 5. Conclusions

This is the first report on changes in the lipid and hydroxycinnamic acid profiles of potato plants after colonization by *B. bassiana.* The main observed effects were increases in the amount of stigmasterol in leaves, hydroxy fatty acid (C16:0 2-OH) in leaves, and hydroxycinnamic acids in roots. Most probably, these changes are associated with oxidative reactions in potato plants during colonization by this *B. bassiana* strain (Sar-31), in agreement with our previous results [13,15]. At the same time, these changes are consistent with plant growth stimulation and the enhancement of mechanisms of defense against phytopathogens [15]. This study expands our knowledge about immunomodulatory and growth-stimulating effects of entomopathogenic fungi during their interaction with plants. Given that the studied strain affects plant biochemistry and stimulates potato growth, there is good potential for the development of a biological product based on this fungus with subsequent commercialization. A future study may also address the effect of *B. bassiana* on the chemical composition of potato tubers because it may affect human health.

## Figures and Tables

**Figure 1 plants-12-03938-f001:**
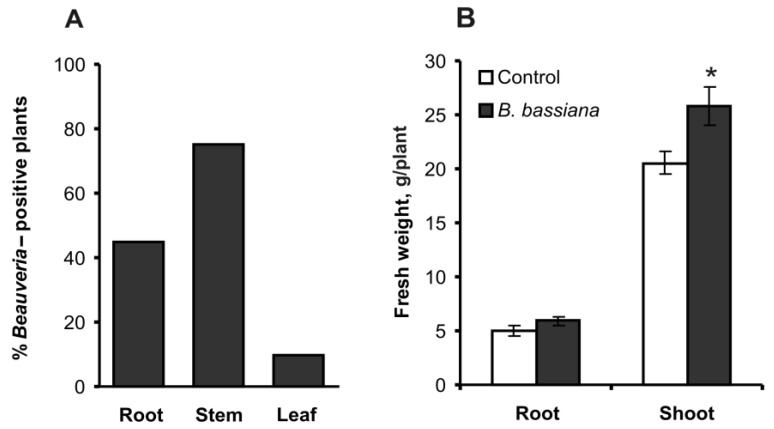
The percentage of *Beauveria*-positive potato plants (**A**) and plant growth parameters (**B**) at 14 days after inoculation of the soil substrate with either *B. bassiana* conidia or an aqueous Tween 80 solution (control); n = 20 per treatment group in each assay. *B. bassiana* was not detectable in the control treatment group. * Significant differences from the control (*t* test, *p* = 0.016, df = 38).

**Figure 2 plants-12-03938-f002:**
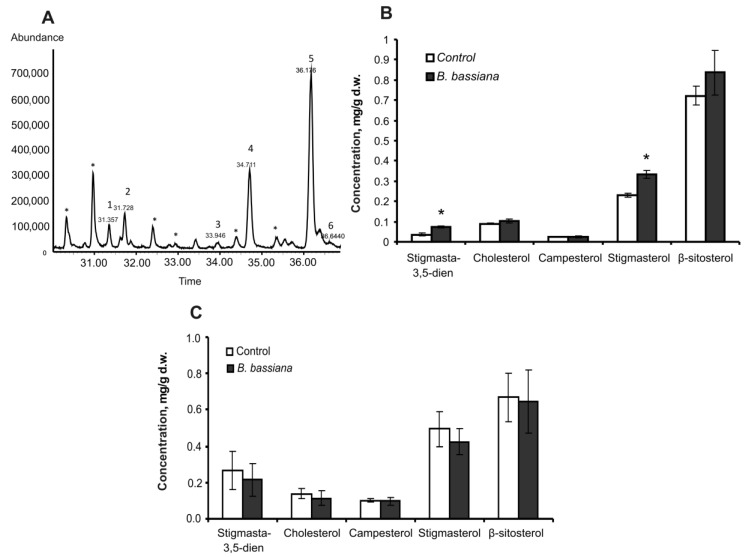
Levels of phytosterols in potato leaves and roots. (**A**) GC-MS analysis of phytosterols in a potato leaf extract. 1: stigmasta-3,5-dien, 2: cholesterol, 3: campesterol, 4: stigmasterol, 5: β-sitosterol, 6: fucosterol. * Linear and branched hydrocarbons C31–C33. (**B**,**C**) Concentrations of phytosterols in leaves (**B**) and roots (**C**) at 14 days after treatment of soil with either *B. bassiana* conidia or the aqueous Tween 80 solution (control); d.w.: dry weight. * Significant differences from the control (*t* test, *p* < 0.005, df = 6).

**Figure 3 plants-12-03938-f003:**
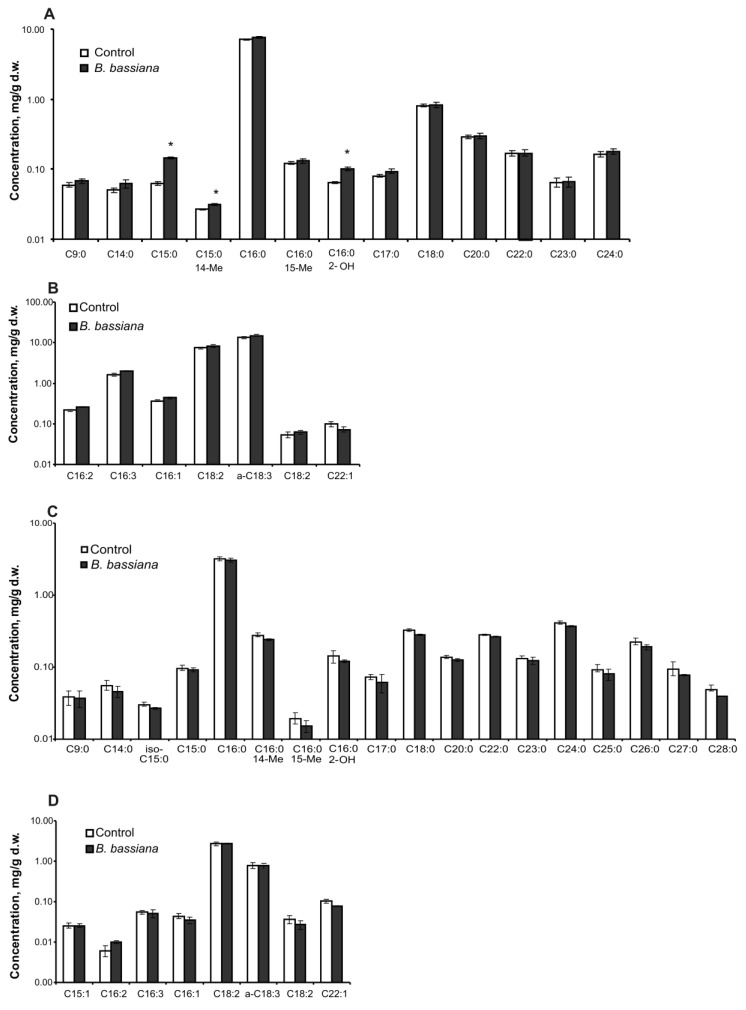
Concentrations of saturated and unsaturated fatty acids in potato leaves (**A**,**B**) and roots (**C**,**D**) at 14 days after treatment of the soil substrate with either *B. bassiana* conidia or the aqueous Tween 80 solution (control); d.w.: dry weight. * *p* < 0.05, as compared to the control (*t* test or Mann–Whitney *U* test performed on four biological replicates).

**Figure 4 plants-12-03938-f004:**
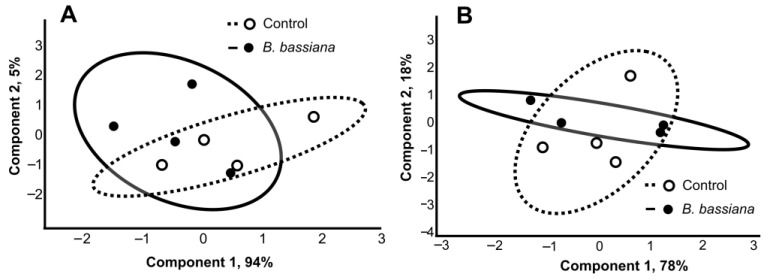
PCA of fatty acid composition of leaves (**A**) and roots (**B**) in potato plants at 14 days after treatment of the soil substrate with either *B. bassiana* conidia or the aqueous Tween 80 solution (control).

**Figure 5 plants-12-03938-f005:**
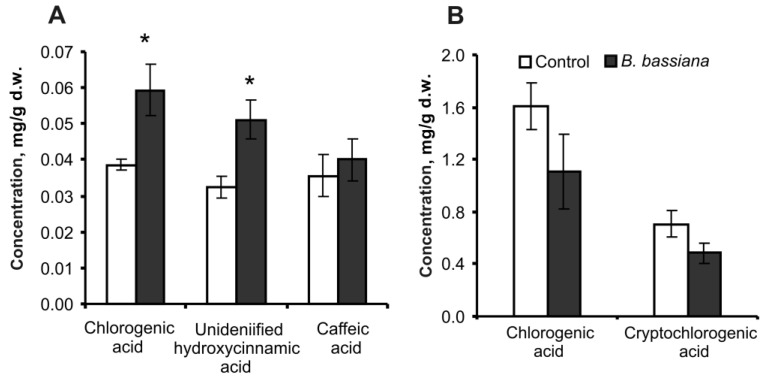
Concentrations of hydroxycinnamic acids in potato roots (**A**) and leaves (**B**) at 14 days after treatment of the soil substrate with either *B. bassiana* conidia or the aqueous Tween 80 solution (control); d.w.: dry weight. * *p* < 0.05 as compared to the control (*t* test or Mann–Whitney *U* test performed on four biological replicates).

## Data Availability

The raw data of this article will be made available by the authors, without restrictions.

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
