# Peer review of "The Fungus Beauveria bassiana Alters Amounts of Sterols, Fatty Acids, and Hydroxycinnamic Acids in Potato Solanum tuberosum"

_plants, 2023, doi:10.3390/plants12233938_

Round 1

Reviewer 1 Report

Comments and Suggestions for Authors

Review opinions

The article shows the changes in levels of sterols, fatty acid, and hydroxycinnamic acids in the roots and leaves of potato plants inoculated by the entomopathogenic endophytic fungus Beauveria bassiana, suggesting that the potential of the endophytic fungus in potato cultivation. In general, the article is good, and I recommend it to be accepted by the journal after it is revised.

Questions about writing

Line 20-21

The sentence should be improved to “….we analyzed the amounts of sterols, fatty acids, and phenolic acids in potato plants inoculated with B. bassiana conidia in soil.”

Line 24

The word “tissues” should be replaced with “plants”, because all the materials used for analysis on the levels of the chemicals were not potato tissues, according to the description in “M and M”.

Line 31-32

A few of references should be necessary.

Line 40

Abiotic stressors should be changed into “abiotic stress” or “environmental stresses”.

Line 54

The word “by” should be changed into “further”

Line 58-59

The related references should be necessary.

Line 67

In this sentence, strictly, fatty acids, and terpenoids don’t belong to lipids

Line 76

The word “stressors” should be changed into “stress”.

Line 108

Please give the full Latin name: Beauveria bassiana, although its name was given before.

Line 113

Please simply introduce how to harvest conidia

Line 117

Please give the full Latin name: Solanum tuberosum, although its name was given before.

Line 119

Why not use the sterilized cultivation substrate?

Line 128

0.03% Tween 80 solution?

Line 152-156

How long did you carry out ultrasonic bath and centrifugation?

And how did you evaporate the organic layer?

Line 159

How about the ratio of 40% methanol and 2% aqueous acetic acid in the mixture?

Line 209

Authors should add a comma before the word “but”.

Line 219-220

Compared with their amounts in roots? If so, please add “compared with their amounts in roots” at the end of the sentence.

Line 240

The underline before caffeic acid should be deleted.

Line 242-245

The sentence should be concise.

Line 248

The word “amount” should be used as “amounts”.

Line 257-259

Authors should point out in which figures the results were shown.

Line 292-293

The related references should be necessary

Line 301-302

Authors should point out in which figures the results were shown.

Line 308-309

The related references should be necessary.

Line 310-311

Authors should point out in which figures the results were shown.

Line 327-328

The sentence is not clear. What meaning is the word “its”? what does the word stand for? the acid or membrane?

Line 332-334

The sentence should be more concise.

In addition, hydroxyl acid (C16:0 2-OH) should be hydroxyl fatty acid.

Questions about Discussion

(1) I think that the discussion about the role of chlorogenic acid in improving plant growth is a little shallow. Authors should research more related references to elucidate its role in improving plant growth.

(2) Some of secondary metabolites shown in the research maybe occur in potato tubers, how about their effect on human health?

Questions about figures

(1) In Fig. 1B, the unit of the y axis should be g/plant.

(2) I suggest that Fig. S1 should be regarded as Fig. 2C. Or these data occurring in Fig. S1 are shown in Fig. 2B.

(3) I suggest that the data shown in Fig. 3C are combined into Fig. 3B. The combined Fig. 3B shows amounts of unsaturated fatty acids and hydroxy fatty acid in potato leaves. Fig. S2 should be regarded as Fig. 3C. Additionally, Fig. S3 and Fig. S4 should be combined together, and regarded as Fig. 3D. Thus, all the data of fatty acids are shown before readers after the article is published.

(4) In Fig. 4, the legend should be shown in Fig. 4B.

(5) Fig. S5 should be regarded as a normal figure shown in the full text. If so, the order of all the figures should be regulated correspondingly.

(6) All the lines in these figures are so thick, and they are a little blinding.

Questions about References

New references, especially those published in 2023, should be cited.

Comments on the Quality of English Language

In general, the writing is good, but some details should be improved, especially questions about classification of metabolites

Author Response

Comment. The article shows the changes in levels of sterols, fatty acid, and hydroxycinnamic acids in the roots and leaves of potato plants inoculated by the entomopathogenic endophytic fungus Beauveria bassiana, suggesting that the potential of the endophytic fungus in potato cultivation. In general, the article is good, and I recommend it to be accepted by the journal after it is revised.

 Response. Thank you for wok with paper and valuable suggestions!

Questions about writing

Comment. Line 20-21 The sentence should be improved to “….we analyzed the amounts of sterols, fatty acids, and phenolic acids in potato plants inoculated with B. bassiana conidia in soil.”

 Response. The sentence was rewritten:  In the present work, gas chromatography–mass spectrometry and high-performance liquid chromatography were used to analyze sterol, fatty-acid, and phenolic-acid concentrations in potato plants inoculated with B. bassiana conidia in soil.

Comment. Line 24 The word “tissues” should be replaced with “plants”, because all the materials used for analysis on the levels of the chemicals were not potato tissues, according to the description in “M and M”.

 Response. The sentence was rewritten:  We propose that these changes could have been caused by oxidative reactions, and the alterations may have resulted in growth-stimulatory and protective effects of B. bassiana on the plants.

Comment. Line 31-32 A few of references should be necessary.

 Response. The paragraph has been modified as requested by reviewer 3 and references added: Entomopathogenic endophytic fungi such as Beauveria and Metarhizium are promising candidates for improving plant health and yield [1-3]. Traditionally, products for the management of herbivores and blood-sucking arthropods have been developed based on their conidia and blastospores. Broad host ranges and the ability to penetrate the cuticle of arthropods gives the fungi an advantage over bacterial and viral biocontrol agents. In the 21st century, it was discovered that these ascomycetes lead a dual lifestyle: aside from parasitism on arthropods, the fungi are facultative symbionts of plants [2]. These fungi are able to colonize roots and aboveground plant organs, thereby stimulating plant growth and immunity and protecting plants against phytopathogens and herbivorous insects [1-2]. Growth- and immune-system-modulating actions of fungi on plants are mediated by several mechanisms including increased nutrient absorption by roots and production of secondary metabolites and hormones eliciting differential regulation of metabolic and immune signaling pathways [4-7]. For example, Raad et al. [6] showed that colonization of Arabidopsis by Beauveria led to reprogramming of metabolic pathways related to resistance to environmental stresses and phytopathogenic microorganisms. Proietti et al. [7] reported differential regulation of proteins associated with defense responses, calcium transport, protein/amino acid turnover, and biosynthesis of energy compounds and hormones in tomato in response to colonization with B. bassiana. Entomopathogenic fungi are able to alter the biochemical composition of plant tissues [8] and attractiveness of plants to phytophagous insects [9].

Comment. Line 40 Abiotic stressors should be changed into “abiotic stress” or “environmental stresses”.

 Response. Corrected

Comment. Line 54 The word “by” should be changed into “further”

 Response. Corrected

Comment. Line 58-59 The related references should be necessary.

 Response. Reference added: https://doi.org/10.3390/horticulturae7080217

Comment. Line 67 In this sentence, strictly, fatty acids, and terpenoids don’t belong to lipids

 Response. According chemical classifications (https://doi.org/10.1016/j.bbalip.2011.06.009) fatty acids and terpenoids are part of lipids. However, due to concerns, we removed the word "lipids" from the sentence

Comment. Line 76 The word “stressors” should be changed into “stress”.

 Response. Corrected

Comment. Line 108 Please give the full Latin name: Beauveria bassiana, although its name was given before.

 Response. Corrected

Comment. Line 113 Please simply introduce how to harvest conidia

 Response. Added: For treatment of plants, conidia were harvested from agar by means of a spatula and dispersed in an aqueous solution of Tween 80 (0.03% v/v).

Comment. Line 117 Please give the full Latin name: Solanum tuberosum, although its name was given before.

 Response. Corrected

Comment. Line 119 Why not use the sterilized cultivation substrate?

 Response. We do not use sterile substrates as this leads to loss of fungistasis and such substrates can often become overgrown with saprotrophic fungi (for example, Trichothecium) during plant cultivation.

Comment. Line 128 0.03% Tween 80 solution?

 Response. Yes. Added.

Comment. Line 152-156 How long did you carry out ultrasonic bath and centrifugation? And how did you evaporate the organic layer?

 Response. Information was added: Water was added to the supernatant, it was centrifuged for 15 min at 2200 g, and the organic layer was separated and evaporated at 40 °C in a water bath.

Comment. Line 159 How about the ratio of 40% methanol and 2% aqueous acetic acid in the mixture?

 Response. Corrected: The plant material was crushed in the ball mill, and phenolic acids were extracted with a mixture of methanol and 2% aqueous acetic acid (2:3 v/v) [53].

Comment. Line 209 Authors should add a comma before the word “but”.

 Response. Added.

Comment. Line 219-220 Compared with their amounts in roots? If so, please add “compared with their amounts in roots” at the end of the sentence.

 Response. Sentence was changed: Exceptions were acids C15:0, C15:0 14Me, and α-hydroxy fatty acid С16:0 2-OH, whose content significantly increased in the leaves of colonized plants compared to control ones (Figure 3A,B).

Comment. Line 240 The underline before caffeic acid should be deleted.

 Response. Corrected

Comment. Line 242-245 The sentence should be concise.

 Response. The sentence was divided into two: The total amount of hydroxycinnamic acids in roots increased 1.5-fold in response to the colonization by B. bassiana (t = 3.1, P = 0.013). In particular, the amount of chlorogenic acid rose by 1.5-fold (Z = 2.2, P = 0.029, Figure 5A). Concentration of the unidentified hydroxycinnamic acid increased by 1.6-fold (t = 3.0, p = 0.022), and the amount of caffeic acid by 1.1-fold (p = 0.40) as compared to the control.

Comment. Line 248 The word “amount” should be used as “amounts”.

 Response. Corrected

Comment. Line 257-259 Authors should point out in which figures the results were shown.

 Response. Added

Comment. Line 292-293 The related references should be necessary

 Response. Explanations added at the request of another reviewer and references provided: The behavior of B. bassiana in a plant is biotrophic according to the classification of Spanu and Kamper [68] because colonization does not induce drastic changes in plant immunity and because tissues remain alive during fungal growth [2].

Comment. Line 301-302 Authors should point out in which figures the results were shown.

 Response. Added

Comment. Line 308-309 The related references should be necessary.

 Response. Added: https://doi.org/10.7717/peerj.9895

Comment. Line 310-311 Authors should point out in which figures the results were shown.

 Response. Added

Comment. Line 327-328 The sentence is not clear. What meaning is the word “its”? what does the word stand for? the acid or membrane?

Response. Corrected: This acid causes membrane permeabilization and disruption of membrane structure, as shown with F. solani and Candida albicans [33,81]

Comment. Line 332-334 The sentence should be more concise. In addition, hydroxyl acid (C16:0 2-OH) should be hydroxyl fatty acid.

Response. Corrected: The main observed effects were increases in the amount of stigmasterol in leaves, hydroxy fatty acid (С16:0 2-OH) in leaves, and hydroxycinnamic acids in roots.

 Questions about Discussion

Comment. (1) I think that the discussion about the role of chlorogenic acid in improving plant growth is a little shallow. Authors should research more related references to elucidate its role in improving plant growth.

Response. The discussion on chlorogenic acid has been expanded:

We observed an elevated amount of hydroxycinnamic acids, especially chlorogenic acid, in potato roots after B. bassiana inoculation (Figure 5). It was shown in a recent work by Muola et al. [72] that B. bassiana colonization of oilseed rape Brassica napus induces biosynthesis of flavonoids including hydroxycinnamic acids and chlorogenic acid. A significant increase of flavonoid concentration in cucumber plants during colonization with B. bassiana (28 days post-inoculation) was also detected by Homayoonzadeh et al. [73]. Researchers [72] hypothesized that mechanisms of this induction may involve a defense response of the plant to B. bassiana or fungal biosynthesis of some compounds; this issue should be elucidated in future studies. It is well known that chlorogenic acid is an antioxidant; therefore, this phenomenon may be associated with a response of potato to oxidative processes. Mei et al. [74] demonstrated that exogenous chlorogenic acid diminishes the levels of H2O2 and malondialdehyde in apple leaves treated with a herbicide inducing oxidative stress. Moreover, exogenous chlorogenic acid modulates peroxidase and catalase production and expression regulation of genes related to the antioxidant system and phenolic-compound metabolism [74]. Those authors believe that chlorogenic acid attenuates membrane damage and lipid oxidation in plants.

Chlorogenic acid plays an important role in processes of plant growth and organogenesis of shoots and roots as well as affects fruit ripening [30,75,76]. The effect of chlorogenic acid may be dose-dependent. In particular, a low concentration of chlorogenic acid accelerates growth of pear tree Pyrus betulaefolia, and a high concentration of the acid can inhibit it [75]. The concentration of chlorogenic acid is closely linked with the amount of a growth-stimulating compound (an auxin called indole-3-acetic acid). This mechanism is mediated by inhibition of indole-3-acetic acid oxidase by phenolic acids, including chlorogenic acid [77,78]. Therefore, upregulation of chlorogenic acid may be one of the reasons for the growth stimulation seen after colonization by B. bassiana. Chlorogenic acid has antifungal properties, in particular, inhibition of spore germination, a reduction in mycelial growth, or cell lysis, which have been documented for Sclerotinia sclerotiorum, Fusarium solani, Verticillium dahliae, Botrytis cinerea, Cercospora sojina [33], Phytophthora infestans [79], and Phlyctaena vagabunda [80]. This acid causes membrane permeabilization and disruption of membrane structure, as shown with F. solani and Candida albicans [33,81]. Moreover, chlorogenic acid is an important intermediate in lignin biosynthesis, and lignification may be one of key mechanisms of disease resistance [32]. Thus, the elevation of the chlorogenic-acid level in potato after the B. bassiana treatment is in agreement with diminished incidence of diseases caused by phytopathogenic fungi [15].

Comment. (2) Some of secondary metabolites shown in the research maybe occur in potato tubers, how about their effect on human health?

Response. Sorry, but the effect of fungi on the chemical composition of tubers is a topic for a separate study. We noted in conclusion that future research could be focused on this topic:  A future study may also address the effect of B. bassiana on chemical composition of potato tubers because it may affect human health.

Questions about figures

Comment. (1) In Fig. 1B, the unit of the y axis should be g/plant.

Response. Done

Comment. (2) I suggest that Fig. S1 should be regarded as Fig. 2C. Or these data occurring in Fig. S1 are shown in Fig. 2B.

Response. Done. Please, see revised manuscript

Comment. (3) I suggest that the data shown in Fig. 3C are combined into Fig. 3B. The combined Fig. 3B shows amounts of unsaturated fatty acids and hydroxy fatty acid in potato leaves. Fig. S2 should be regarded as Fig. 3C. Additionally, Fig. S3 and Fig. S4 should be combined together, and regarded as Fig. 3D. Thus, all the data of fatty acids are shown before readers after the article is published.

Response. Done as required

Comment. (4) In Fig. 4, the legend should be shown in Fig. 4B.

Response. Done

Comment. (5) Fig. S5 should be regarded as a normal figure shown in the full text. If so, the order of all the figures should be regulated correspondingly.

Response. Done

Comment. (6) All the lines in these figures are so thick, and they are a little blinding.

Response. Lines reduced from 0.6 to 0.4 mm

Questions about References

Comment. New references, especially those published in 2023, should be cited.

Response. New references (2022-2023) have been added:

Quesada-Moraga; Garrido-Jurado, I.; González-Mas, N.;  Yousef-Yousef, M. Ecosystem services of entomopathogenic ascomycetes, J. Invertebr. Pathol. 2023, 201, 108015, https://doi.org/10.1016/j.jip.2023.108015.

Rasool, S.; Markou, A.; Hannula, S.E.;  Biere, A. Effects of tomato inoculation with the entomopathogenic fungus Metarhizium brunneum on spider mite resistance and the rhizosphere microbial community. Front. Microbiol. 2023, 14, 1197770. https://doi.org/10.3389/fmicb.2023.1197770

Iida, Y.; Higashi, Y.; Nishi, O.; Kouda, M.; Maeda, K.; Yoshida, K.; Asano, S.; Kawakami, T.; Nakajima, K.; Kuroda, K.; Tanaka, C.; Sasaki, A.; Kamiya, K.; Yamagishi, N.; Fujinaga, M.; Terami, F.; Yamanaka, S.; Kubota, M. Entomopathogenic fungus Beauveria bassiana-based bioinsecticide suppresses severity of powdery mildews of vegetables by inducing the plant defense responses. Front. Plant Sci. 2023, 14, 1211825. https://doi.org/10.3389/fpls.2023.1211825

Muola, A.; Birge, T.; Helander, M.; Mathew, S.; Harazinova, V.; Saikkonen, K.; Fuchs B. Endophytic Beauveria bassiana induces biosynthesis of flavonoids in oilseed rape following both seed inoculation and natural colonization. Pest Manag. Sci. 2023, https://doi.org/10.1002/ps.7672

Homayoonzadeh, M.; Esmaeily, M.; Talebi, K.; Allahyari, H.; Reitz, S.; Michaud, J.P. Inoculation of cucumber plants with Beauveria bassiana enhances resistance to Aphis gossypii (Hemiptera: Aphididae) and increases aphid susceptibility to pirimicarb. Eur. J. Entomol. 2022, 119, 1-11. https://doi.org/10.14411/eje.2022.001

Yang, Y.; Cui, S.; Zhang, Y.; Wang, X.; Li, D.; Wang, R. PbHCT4 regulates growth through affecting chlorogenic acid (cga) content in pear. Sci. Hortic. 2022, 303, 111225. https://doi.org/10.1016/j.scienta.2022.111225

Comments on the Quality of English Language

Comment. In general, the writing is good, but some details should be improved, especially questions about classification of metabolites.

Response. Additional proofreading has been performed (http://shevchuk-editing.com/). The chemical classification of metabolites is given according to Fahy et al. (2011) "Lipid classification, structures and tools" (https://doi.org/10.1016/j.bbalip.2011.06.009)

Reviewer 2 Report

Comments and Suggestions for Authors

The authors in this study intend to assess the possible physiological changes by inoculating the Beauveria bassiana fungi in potato plants. The sterol, fatty-acid, and phenolic-acid amounts in potato was analyzed using different analytical techniques. The results exhibit significant increase in the stigmasterol, minor sterol compounds, and some hydroxy fatty acids in leaves and hydroxycinnamic acids in roots with fungal inoculation. This is an intersting study and is in the scope of journal but need attention of authors – there are some grammatically weak sentences so it is better the ms should be read by English native colleague for fluency in language.

This study is not well-designed and I have some reservations in Methodology section which is confusing and need more clarity. The authors did not give relevant citations to most of the protocols in Methodology section and overall this section is weakly written/explained. There is need to substantially improve this section with most relevant/authentic citations (especially sub-section 2.1, 2.2 and 2.3). The time of neither material collections nor experiments is given throughout the ms, how the assays were conducted etc., should be given in more detail and supported with relevant citation/s? What type of selective media was used to grow fungi without contamination, the most important concern is the exact ID of endophytic B. bassiana (there are more specific media for isolated fungi – why would they not be used? Doberski & Tribe media for Beauveria for example.). The confirmation of the status of endophytic fungi in potato plants (P3, L140) is not possible without molecular tools, this is very important to correctly identify that the colonized fungi matches with the treated fungi? – additional experiments should be included in the ms for this confirmation.

I am of the opinion that in the current form this ms can not be accepted for its publication in Plants but based on the novelty and importance of study the Editor may give an opportunity to authors to substantially revise the article for consideration.

Some of my comments are as follows:

The authors should consider to give citation of disinfestation process?, explain the protocol of inoculation of fungi in potato and provide relevant citation/s? provide dimensions of pots/vases and how was the endophytic fungus verified to be the treatment fungus? In my opinion this is the main constraint of this study as molecular verification of the fungus as an endophyte (see Ramakuwela et al. 2019, https://doi.org/10.1016/j.biocontrol.2019.104102 and others) is important – the possibility of contamination cannot be overruled without the exact ID of the endophytic status?

P1, L21, “replace word “registered” with appropriate word

P3, L108, what was the source of Sar-31?

P3, L112, why 0.25% of yeast was supplemented, give citation as normally there is 0.01% is added?

P3, L115, what was the percentage of viability of conidia?

P3, L126, how much was the concentration of Tween 80 both in treated and control pots?

P4, L186, give citation for Levene’s test?

The Conclusion of study should be give at the end of Discussion section

It is suggested authors should carefully check all the citations with corresponding list of references in the ms

Comments on the Quality of English Language

There are some grammatically weak sentences so it is better the ms should be read by English native colleague for fluency in language

Author Response

Comment. The authors in this study intend to assess the possible physiological changes by inoculating the Beauveria bassiana fungi in potato plants. The sterol, fatty-acid, and phenolic-acid amounts in potato was analyzed using different analytical techniques. The results exhibit significant increase in the stigmasterol, minor sterol compounds, and some hydroxy fatty acids in leaves and hydroxycinnamic acids in roots with fungal inoculation. This is an intersting study and is in the scope of journal but need attention of authors – there are some grammatically weak sentences so it is better the ms should be read by English native colleague for fluency in language.

This study is not well-designed and I have some reservations in Methodology section which is confusing and need more clarity. The authors did not give relevant citations to most of the protocols in Methodology section and overall this section is weakly written/explained. There is need to substantially improve this section with most relevant/authentic citations (especially sub-section 2.1, 2.2 and 2.3). The time of neither material collections nor experiments is given throughout the ms, how the assays were conducted etc., should be given in more detail and supported with relevant citation/s?

Response. Thank you for work with paper and valuable comments. We performed additional assays and tried to correct all issues. Sub-section 2.1, 2.2 and 2.3 were supplemented and citations were added. Now the subsections look as:

2.1. The fungus and its cultivation

An isolate of Beauveria bassiana s.s. (strain Sar-31) from the collection of microorganisms at the Institute of Systematics and Ecology of Animals (the Siberian Branch of the Russian Academy of Sciences) was used. The strain was isolated from Calliptamus italicus (Orthoptera, Acrididae) in Western Siberia (53°41′N 78°02′E) in 2001. The fungal species was identified on the basis of the sequence of a translation elongation factor region (GenBank accession No. MZ564259). The fungus was grown in Petri dishes (90 mm) with 1/4 Sabouraud dextrose agar supplemented with 0.25% of yeast extract (1/4 SDAY [35]) at 26 °C in the dark for 10 days. For treatment of plants, conidia were harvested from agar by means of a spatula and dispersed in an aqueous solution of Tween 80 (0.03% v/v). The concentration of conidia was determined using a hemocytometer. Germination of conidia on 1/4 SDAY was 98% after 16 h of incubation (n = 100).

2.2. Plant cultivation and inoculation with the fungus

Potato plants (Solanum tuberosum variety Agata) were used in the experiments. Tubers (diameter 3–4 cm) were treated with 7% sodium hypochlorite (2 min) and then 70% ethanol, followed by three rinses in sterile distilled water [12]. The tubers were pre-exposed to light at 25 °C for 30 days for initiation of germination (shoot length ~3 mm) [12]. Germinated tubers were planted individually in cone-shaped containers (1 L, height: 12 cm, lower diagonal: 10 cm, upper diagonal: 14 cm) with 300 g of unsterilized soil substrate Terra Vita (Nord Palp, Fornosovo, Russia). The substrate includes peat, purified sand, perlite, mineral micro- and macro-elements, and has a pH of 6.5. The substrate was analyzed for the presence of fungi Metarhizium and Beauveria by plating on CTAB medium [36,37] with a minor modification [14,15], and these fungi were not detectable.

Plants were grown at 22°C and 40–60% relative humidity with a 16 h/8 h photoperiod (light/dark) using full-spectrum phytolamps Flora (Osram, Munich, Germany). We used soil drenching with the conidial suspension around seedlings ([38-40] with modifications) because preliminary assays showed effective colonization of potatoes under the laboratory conditions described above. Soil around tubers was inoculated with 10 mL of the suspension (108 conidia per mL) in the 0.03% (v/v) Tween 80 solution at 3 days after planting. Control plants were treated with a conidia-free aqueous Tween 80 solution.

2.3. The extent of colonization and plant weight

Time point “14 days after treatment” was selected for analysis. This choice is due to preliminary assays’ showing that this period is sufficient for implementing colonization of plant tissues by the fungal strain. Plants were excavated and carefully cleansed from the soil substrate. Fresh weights of roots and shoots were determined with a precision of 0.01 g. Endophytic colonization of the plants was estimated according to Posada et al. [41]. Roots, stems, and leaves were surface sterilized by means of 0.5% sodium hypochlorite for 2 min and 70% ethanol for 2 min with subsequent 3-fold washing in sterile distilled water. Then, the plant organs were plated in 90 mm Petri dishes (two fragments of a root or stem or two leaves per dish) on the CTAB medium [36-37] with a modification (10 g/L peptone, 40 g/L D-glucose, 20 g/L agar, 1 g/L yeast extract, 0.35 g/L cetyltrimethylammonium bromide, 0.05 g/L cycloheximide, 0.05 g/L tetracycline, and 0.6 g/L streptomycin). For verification of surface sterilization, tissue imprinting onto this medium was performed [42]. The dishes were incubated at 24 °Ð¡ for 12 days, and fungal growth was examined visually. The colonies were identified as Beauveria by light microscopy. Twenty plants per treatment group were used for the determination of weight and colonization frequency.

References from these subsections:

  1. Krell, V.; Unger, S.; Jakobs-Schoenwandt, D.; Patel, A.V. Endophytic Metarhizium brunneum mitigates nutrient deficits in potato and improves plant productivity and vitality. Fungal Ecol. 2018, 34, 43–49. https://doi.org/10.1016/j.funeco.2018.04.002
  2. Tyurin, M.; Kabilov, M.R.; Smirnova, N.; Tomilova, O.G.; Yaroslavtseva, O.; Alikina, T.; Glupov, V.V.; Kryukov, V.Y. Can Potato Plants Be Colonized with the Fungi Metarhizium and Beauveria under Their Natural Load in Agrosystems? Microorganisms 2021, 9, 1373. https://doi.org/10.3390/microorganisms9071373
  3. Tomilova, O.G.; Shaldyaeva, E.M.; Kryukova, N.A.; Pilipova, Y.V.; Schmidt, N.S.; Danilov, V.P.; Kryukov, V.Y.; Glupov, V.V. Entomopathogenic fungi decrease Rhizoctonia disease in potato in field conditions. PeerJ 2020, 8, e9895 https://doi.org/10.7717/peerj.9895
  4. Bischoff, J.F.; Rehner, S.A.; Humber, R.A. A multilocus phylogeny of the Metarhizium anisopliae lineage. Mycologia 2009, 101, 512–530. https://doi.org/10.3852/07-202
  5. Posadas, J.B.; Comerio, R.M.; Mini, J.I.; Nussenbaum, A.L., Lecuona, R.E. A novel dodine-free selective medium based on the use of cetyl trimethyl ammonium bromide (CTAB) to isolate Beauveria bassiana, Metarhizium anisopliae sensu lato and Paecilomyces lilacinus from soil. Mycologia 2012, 104, 974–980. https://doi.org/10.3852/11-234
  6. Kepler, R.M.; Ugine, T.A.; Maul, J.E.; Cavigelli, M.A.; Rehner, S.A. Community composition and population genetics of insect pathogenic fungi in the genus Metarhizium from soils of a long-term agricultural research system. Environ. Microbiol. 2015, 17, 2791–2804. https://doi.org/10.1111/1462-2920.12778
  7. Rasool, S.; Markou, A.; Hannula, S.E.; Biere, A. Effects of tomato inoculation with the entomopathogenic fungus Metarhizium brunneumon spider mite resistance and the rhizosphere microbial community. Front. Microbiol. 2023, 14, 1197770. https://doi.org/10.3389/fmicb.2023.1197770
  8. Siqueira, A.C.O.; Mascarin, G.M.; Gonçalves, C.R.N.C.B.; Marcon, J.; Quecine, M.C.; Figueira, A.; Delalibera, Í.Jr. Multi-Trait Biochemical Features of Metarhizium Species and Their Activities That Stimulate the Growth of Tomato Plants. Front. Sustain. Food Syst. 2020, 4, 137. doi: 10.3389/fsufs.2020.00137
  9. Baron, N.C.; Souza, Pollo, Ad.; Rigobelo, E.C. Purpureocillium lilacinum and Metarhizium marquandii as plant growth-promoting fungi. PeerJ 2020, 8:e9005 https://doi.org/10.7717/peerj.9005
  10. Posada, F.; Aime, M.C.; Peterson, S.W.; Rehner, S.A.; Vega, F.E. Inoculation of coffee plants with the fungal entomopathogen Beauveria bassiana (Ascomycota:Hypocreales). Mycological Research 2007, 111, 748-757. https://doi.org/10.1016/j.mycres.2007.03.006.
  11. 42. McKinnon, A.C.; Saari, S.; Moran-Diez, M.E.; Meyling, N.V.; Raad, M.; Glare, T.R. Beauveria bassiana as an endophyte: a critical review on associated methodology and biocontrol potential. BioControl 2017, 62, 1-17 https://doi.org/10.1007/s10526-016-9769-5

Comment. What type of selective media was used to grow fungi without contamination, the most important concern is the exact ID of endophytic B. bassiana (there are more specific media for isolated fungi – why would they not be used? Doberski & Tribe media for Beauveria for example.). The confirmation of the status of endophytic fungi in potato plants (P3, L140) is not possible without molecular tools, this is very important to correctly identify that the colonized fungi matches with the treated fungi? – additional experiments should be included in the ms for this confirmation. I am of the opinion that in the current form this ms can not be accepted for its publication in Plants but based on the novelty and importance of study the Editor may give an opportunity to authors to substantially revise the article for consideration.

Response. We used half-strength CTAB (cetyltrimethylammonium bromide) medium (Kepler et al 2016, DOI: 10.1111/1462-2920.12778) which also selective for Beauveria and Metarhizium (10.1111/1462-2920.12778, DOI: 10.3852/11-234). Explanations and references added in text.

Regarding molecular identification, we performed additional tests to identify the isolated re-isolates of B. bassiana. Corresponded subsecions and one table in ESM were added:

Methods:

2.4. PCR-based confirmation of reisolate identity

To confirm the identity of endophytic fungi reisolated from plants exposed to the initial B. bassiana strain Sar-31, genomic DNA extraction was followed by PCR with specific primers and amplicon purification and sequencing. For precise discrimination of fungal strain, which may have homologous sequences of a translation elongation factor and other widely used genes [43,44], an approach of Levchenko et al. [45] was applied, utilizing another, more variable locus. Primers slipF1 (5¢-GTGGAAGCTCGCCGAGAG-3¢) and slipR1 (5¢-GCAGATGSACCTCGTTGC-3¢) were used, which flank a ~500 bp region of the gene of secreted lipase: a protein from the superfamily of a/β-hydrolases. Five additional strains of B. bassiana from the collection of microorganisms at the All-Russian Institute of Plant Protection were chosen for sequence variation comparison (Table S1).

A simplified protocol of DNA extraction [46] was applied to the samples of fungal mycelia grown in pure culture. PCR products were separated by gel electrophoresis, and bands of specific size were excised, gel-purified according to Vogelstein and Gillespie [47], and sequenced in both directions on an ABI Prism sequencer (Thermo Fisher Scientific, Waltham, Massachusetts, USA). The obtained sequences were corrected manually and aligned in BioEdit [48].

Results:

3.1. Colonization and growth parameters

The extent of colonization of internal plant tissues by B. bassiana at 14 days postinoculation was 45% for roots, 75% for stems, and 10% for leaves (Figure 1a). No B. bassiana–colonized tissues were found in the control treatment group. Cultures reisolated from plants showed 100% identity of the secreted-lipase gene fragment (445 bp) toward each other and to the paternal strain Sar-31 (Table S1). On the other hand, the fragment was variable enough to discriminate different B. bassiana strains, including those originating from the same location, namely Sar-31 and BBK-1 (Table S1). Thus, potential contamination of a particular culture of Beauveria with a random fungal strain is highly likely to be tracked down by genotyping, showing a different sequence. This allows to presume that there was no contamination during the study and that all the cultures used for the experiment correspond to the paternal strain Sar-31.

Comment. Some of my comments are as follows:The authors should consider to give citation of disinfestation process?, explain the protocol of inoculation of fungi in potato and provide relevant citation/s? provide dimensions of pots/vases and how was the endophytic fungus verified to be the treatment fungus? In my opinion this is the main constraint of this study as molecular verification of the fungus as an endophyte (see Ramakuwela et al. 2019, https://doi.org/10.1016/j.biocontrol.2019.104102 and others) is important – the possibility of contamination cannot be overruled without the exact ID of the endophytic status?

Response. Please, see responses above.

Comment. P1, L21, “replace word “registered” with appropriate word

Response. Changed on "showed"

Comment. P3, L108, what was the source of Sar-31?

Response. Added: The strain was isolated from Calliptamus italicus (Orthoptera, Acrididae) in Western Siberia (53°41′N 78°02′E) in 2001.

Comment. P3, L112, why 0.25% of yeast was supplemented, give citation as normally there is 0.01% is added?

Response. These amount may be different: from 0.01% to 1% depending on fungal species and strain (for example, 10.1016/j.funbio.2017.07.007). We used one quarter SDA with 0.25% yeast  extract because it is optimal for strain Sar 31 (many conidia, few mycelium). Reference added. 

Comment. P3, L115, what was the percentage of viability of conidia?

Response. Information was added:  Germination of conidia on 1/4 SDAY was 98% after 16 h of incubation (n = 100).

Comment. P3, L126, how much was the concentration of Tween 80 both in treated and control pots?

Response. added: 0.03% (v/v)

Comment. P4, L186, give citation for Levene’s test?

Response. Added: Levene, H. Robust tests for equality of variances. In  Contributions to Probability and Statistics I. Olkin Ed. Stanford University Press: Palo Alto, USA, 1960. pp. 278–292.

Comment. The Conclusion of study should be give at the end of Discussion section

Response. The conclusion is expanded and looks as:

In conclusion, this is the first report on changes in lipid and hydroxycinnamic-acid profiles of potato plants after colonization by B. bassiana. The main observed effects were increases in the amount of stigmasterol in leaves, hydroxy fatty acid (С16:0 2-OH) in leaves, and hydroxycinnamic acids in roots. Most probably, these changes are associated with oxidative reactions in potato plants during colonization by this B. bassiana strain (Sar-31), in agreement with our previous results [13,15]. At the same time, these changes are consistent with plant growth stimulation and enhancement of mechanisms of defense against phytopathogens [15]. This study expands our knowledge about immunomodulatory and growth-stimulating effects of entomopathogenic fungi during their interaction with plants. Given that the studied strain affects plant biochemistry and stimulates potato growth, there is good potential for the development of a biological product based on this fungus with subsequent commercialization. A future study may also address the effect of B. bassiana on chemical composition of potato tubers because it may affect human health.

Comment. It is suggested authors should carefully check all the citations with corresponding list of references in the ms

Response. All citations carefully checked and 20 new references were added

Comment. Comments on the Quality of English Language. There are some grammatically weak sentences so it is better the ms should be read by English native colleague for fluency in language

Response. Professional proofreading was performed (http://shevchuk-editing.com/).We would appreciate any further suggestions to improve the manuscript.

Reviewer 3 Report

Comments and Suggestions for Authors

This is an interesting study of the influence of an entomopathogenic fungus (Beauveria bassiana) following the colonization of potato plants, looking at growth factors and levels of sterols, fatty acids, and hydroxycinnamic acids in leaves, stems, and roots. In this case, Beauveria was considered to be acting as an endophyte.

The introduction should explain more clearly the role of entomopathogenic fungi in biological control of agricultural pests.

There was no mention of potato pests that endophytic Beauveria could protect the plants from, or potato diseases. No mention of the possible use of Beauveria in the field or other possible types of inoculation techniques.

The methodology is well described but it might have been interesting to test the plants at different times after inoculation with the fungus, rather than just one time point (14 days). Many other studies have been carried out using 3-4 time points, at least for plant growth parameters. I can understand that would generate a lot of data to analyze via chromatography etc., but would greatly enrich the study.

There is no mention of “N” or “n” for the endophytic colonization or plant growth factors! This is very important information.

The authors need to be careful when making statements about, for example, protection gained from endophytic colonization against plant pathogens or pests when they have no data on these aspects. I have highlighted the PDF where this was not appropriate or the text appeared to indicate that the authors have this type of data.

The importance of the current findings for commercial use is not discussed and the fact that root growth was not significantly improved with this treatment was a negative aspect, however, this could have been due to the short time period used here. Especially as the tuber (root) is the part of the plant that is commercialized.  

Comments on the Quality of English Language

I have made some suggestions for improving the English but some of the constructs can be further improved. I would suggest asking a native speaker who understands the area of the research to correct this.

Author Response

Comment. This is an interesting study of the influence of an entomopathogenic fungus (Beauveria bassiana) following the colonization of potato plants, looking at growth factors and levels of sterols, fatty acids, and hydroxycinnamic acids in leaves, stems, and roots. In this case, Beauveria was considered to be acting as an endophyte.

Response. Thank you for wok with paper and valuable suggestions!

Comment. The introduction should explain more clearly the role of entomopathogenic fungi in biological control of agricultural pests.

Response. Text of first paragraph was modified:

Entomopathogenic endophytic fungi such as Beauveria and Metarhizium are promising candidates for improving plant health and yield [1-3]. Traditionally, products for the management of herbivores and blood-sucking arthropods have been developed based on their conidia and blastospores. Broad host ranges and the ability to penetrate the cuticle of arthropods gives the fungi an advantage over bacterial and viral biocontrol agents. In the 21st century, it was discovered that these ascomycetes lead a dual lifestyle: aside from parasitism on arthropods, the fungi are facultative symbionts of plants [2]. These fungi are able to colonize roots and aboveground plant organs, thereby stimulating plant growth and immunity and protecting plants against phytopathogens and herbivorous insects [1-2]. Growth- and immune-system-modulating actions of fungi on plants are mediated by several mechanisms including increased nutrient absorption by roots and production of secondary metabolites and hormones eliciting differential regulation of metabolic and immune signaling pathways [4-7]. For example, Raad et al. [6] showed that colonization of Arabidopsis by Beauveria led to reprogramming of metabolic pathways related to resistance to environmental stresses and phytopathogenic microorganisms. Proietti et al. [7] reported differential regulation of proteins associated with defense responses, calcium transport, protein/amino acid turnover, and biosynthesis of energy compounds and hormones in tomato in response to colonization with B. bassiana. Entomopathogenic fungi are able to alter the biochemical composition of plant tissues [8] and attractiveness of plants to phytophagous insects [9].

Comment. There was no mention of potato pests that endophytic Beauveria could protect the plants from, or potato diseases. No mention of the possible use of Beauveria in the field or other possible types of inoculation techniques.

Response.  We try to write an abstract according to general principles: relevance, methods, results, a short conclusion representing a synthesis with previous knowledge. Growth stimulation of potato and inhibition of phytopathogens in field conditions has been shown in previous works (10.7717/peerj.9895). However, an abstract does not imply references. Therefore, we lowered the tone of the first sentence by removing the potatoes from there. It now looks like this: The entomopathogenic endophytic fungus Beauveria bassiana can colonize plants resulting in growth promotion and protection against phytopathogenic microorganisms.

Comment. The methodology is well described but it might have been interesting to test the plants at different times after inoculation with the fungus, rather than just one time point (14 days). Many other studies have been carried out using 3-4 time points, at least for plant growth parameters. I can understand that would generate a lot of data to analyze via chromatography etc., but would greatly enrich the study.

Response. The choice of time point was determined on the basis of preliminary experiments assessing plant colonization by the fungus. A two-week period is enough to obtain at least 50% of the plants with the presence of the fungus in the plant tissues. In addition, potato plants are quite massive and can experience stress when grown in pots for a long time. Unfortunately, our capacity to incubate large numbers of plants and use gas chromatography did not allow earlier time points. Some explanations are included in section 2.3. Please see revised version

Comment. There is no mention of “N” or “n” for the endophytic colonization or plant growth factors! This is very important information.

Response. Please, see section 2.3., last sentence: "Twenty plants per treatment group were used for the determination of weight and colonization frequency". We also added information about n per treatment and df in figure legend.

Comment. The authors need to be careful when making statements about, for example, protection gained from endophytic colonization against plant pathogens or pests when they have no data on these aspects. I have highlighted the PDF where this was not appropriate or the text appeared to indicate that the authors have this type of data.

Response. The protection of potato from phytopathogens during colonization by the B. bassiana has been described in previous work (10.7717/peerj.9895). We carefully re-read the text and added references where they were missed. We are very grateful for corrections in the PDF file and most of them have been reflected. Please, see the revised version in tracking regime. Notably, a few of your corrections have been corrected again by our English editor. They are highlighted in blue in the text.

Comment. The importance of the current findings for commercial use is not discussed and the fact that root growth was not significantly improved with this treatment was a negative aspect, however, this could have been due to the short time period used here. Especially as the tuber (root) is the part of the plant that is commercialized.  

Response. Changes in tuber yield and its quality were described in previous paper (10.7717/peerj.9895). We indicated this in introduction section: " preplanting treatment of potato tubers with either B. bassiana or M. robertsii reduced the incidence of rhizoctoniosis (Rhizoctonia solani), changed immune-activity parameters, and improved crop quality under field conditions [15]. In particular B. bassiana (strain Sar-31) significantly decreased Rhizoctonia disease and improved the tuber yield [15]". We also added in conclusion: ". Given that the studied strain affects plant biochemistry and stimulates potato growth, there is good potential for the development of a biological product based on this fungus with subsequent commercialization."

Comment. I have made some suggestions for improving the English but some of the constructs can be further improved. I would suggest asking a native speaker who understands the area of the research to correct this

Response. Proffessional english editing has been performed (http://shevchuk-editing.com/)

Round 2

Reviewer 2 Report

Comments and Suggestions for Authors

The authors substantially improved their manuscript and added some more tests, I am of the opinion this article may be accepted for its publication in Plants

Comments on the Quality of English Language

Minor spell and grammar check is required which I believe will be done at copy-editing stage!